# Distinct Rabi splitting in confined systems of MoSe$_2$ monolayers and (Ga,In)As quantum wells

Felix Schäfer [1,7], Henry Mittenzwey[2,7], Markus Stein [1 ✉], Oliver Voigt[2], Lara Greten [2], Daniel Anders[1], Isabel Müller[1], Florian Dobener [1], Marzia Cuccu[3], Christian Fuchs[4], Kenji Watanabe [5], Takashi Taniguchi [6], Alexey Chernikov [3], Kerstin Volz[4], Andreas Knorr [2] & Sangam Chatterjee [1]

Rabi splitting is a defining signature of strong light-matter interaction, emerging when a two-level system is resonantly driven by an optical field, resulting in a spectral doublet separated by the Rabi energy. In solid-state systems, Rabi splitting occurs at exciton resonances, where it is shaped by many-body interactions intrinsic to the material. Here, we investigate the Rabi splitting dynamics in two paradigmatic two-dimensional semiconductors: a hBN-encapsulated MoSe$_2$ monolayer and a (Ga,In)As multiple quantum well structure. In MoSe$_2$, strong Coulomb interactions dominate over light-matter coupling, while in the quantum wells, both interactions are of comparable strength. While both systems exhibit clear Rabi splitting under resonant excitation, their behavior diverges under increased excitation strength. MoSe$_2$ displays sublinear Rabi splitting due to excitonic correlations, whereas (Ga,In) As quantum wells reveal additional spectral resonances and coherent optical gain, indicating a transition beyond the simple two-level regime. These contrasting behaviors are quantitatively captured by a unified microscopic many-body theory based on Heisenberg equations of motion and an exciton expansion. Our findings elucidate the impact of many-body interactions on coherent exciton dynamics and establish a framework for tailoring strong-field optical responses in two-dimensional materials.

The coherent nonlinear interaction of light and matter at the quantum level unveils rich dynamics with fascinating effects across various fields of research[1-4]: from atomic[5] and molecular physics[6] to condensed matter systems[7] and quantum optics[8,9]. The observation of level-splitting phenomena associated with electronic transitions dressed by photons is among the most fundamental. For example, the interaction of photons in resonance with the transition energy of a quantum two-level system (2-LS) results in the splitting of its absorption into distinct sidebands. Such characteristic spectral manifestations are termed Autler-Townes splitting, dynamical Stark

[1]Institute of Experimental Physics I and Center for Materials Research (LaMa), Justus-Liebig-University Giessen, Heinrich-Buff-Ring 16, D-35392 Giessen, Germany. [2]Nichtlineare Optik und Quantenelektronik, Institut für Physik und Astronomie (IFPA), Technische Universität Berlin, D-10623 Berlin, Germany. [3]Dresden Integrated Center for Applied Physics and Photonic Materials (IAPP) and Würzburg-Dresden Cluster of Excellence ct.qmat, Technische Universität Dresden, D-01062 Dresden, Germany. [4]Structure & Technology Research Laboratory (WZMW), Philipps-University Marburg, Hans-Meerwein-Straße 6, D-35032 Marburg, Germany. [5]Research Center for Electronic and Optical Materials, National Institute for Materials Science, 1-1 Namiki, Tsukuba 305-0044, Japan. [6]Research Center for Materials Nanoarchitectonics, National Institute for Materials Science, 1-1 Namiki, Tsukuba 305-0044, Japan. [7]These authors contributed equally: Felix Schäfer, Henry Mittenzwey. ✉e-mail: markus.stein@exp1.physik.uni-giessen.de

splitting or Rabi splitting. Originally reported for optically driven molecules in the gas phase[10], these spectral splittings have been attributed to the atomic states being "dressed" by the light field. The emerging states are then separated by the Rabi energy, which scales with the transition dipole moment and the driving field amplitude. The emergence of sufficiently sharp optical transitions due to improved material quality enables similar research on condensed matter systems[11–13]. Notably, these are inherently Coulomb-interacting many-body systems. They natively exhibit more intricate dynamics due to electron-hole collisions and excitation-induced dephasing[14–19]. By tailoring these interactions—e.g., introducing free charges, applying strain gradients, or modifying the environment—such effects can be harnessed to enhance device performance. Unfortunately, such many-body effects also render the observation of splitting phenomena challenging. Consequently, observations of Rabi oscillations in condensed matter systems are relatively rare[20–23]. However, theoretical studies predict that light-dressing can significantly reduce the Coulomb-collision rates[24], offering pathways for relaxation rate control in devices. Experimental studies of Rabi splitting of exciton resonances within two-dimensional (2D) charge carrier systems can directly confirm these predictions. Additionally, amplifying the resonant optical excitation access nonperturbative excitation regimes, leading to the emergence of new absorption resonances and coherent gain phenomena beyond Rabi splitting[25–28]. High-quality transition metal dichalcogenides (TMDCs) or quantum well (QW) structures serve as ideal platforms for studying these phenomena in 2D many-body systems. Such studies transcend earlier investigations of Rabi splitting in microcavity systems, where light-matter interactions are enhanced at exciton resonances[29–31], as well as at intersubband transitions[28,32]. More recently, Rabi splitting has been observed in TMDC monolayers across various cavity designs[33–37]. Beyond resonant optical excitation, Rabi splitting of the 1s exciton also occurs when driven with lower energies in the range of the exciton binding energies[38–42].

In this work, we demonstrate and compare the Rabi splitting dynamics of exciton resonances in two prototypical 2D semiconductor heterostructures—a TMDC single monolayer and conventional (Ga,In) As QWs—under controlled excitation conditions at cryogenic temperatures of 6 K. These systems represent two fundamentally different regimes of light-matter interaction, allowing us to disentangle how the interplay between Coulomb interactions and optical driving governs Rabi dynamics in condensed matter systems. The key distinction lies in the vastly different exciton binding energies relative to the observed Rabi energies in each system, as well as in the applied excitation schemes. These parameters determine the physical nature and fundamental processes as identified by a rigorous microscopic many-

body theory: on the one hand, the TMDC monolayer with strongly bound excitons in co-linear excitation resulting in a multitude of many-particle Coulomb correlations represents the one pole, where Coulomb interaction dominates completely over the light-matter interaction. On the other hand, the (Ga,In)As QW with less strongly bound excitons in co-circular excitation, where only a fraction of Coulomb correlations are possible, represents the other pole, where Coulomb and light-matter interaction are of similar magnitude. This contrast allows us to explore and interpret the fundamentally different nonlinear features observed in each system such as sublinear splitting, spectral sidebands, and coherent gain within a unified theoretical framework.

## Results

TMDCs featuring large exciton binding energies, strong light-matter coupling accompanied by comparatively narrow linewidths are among the clearest realizations of an interacting two-level exciton gas. In particular, MoSe₂ exhibits a singular, bright 1s exciton as the lowest energy transition with binding energies of >100 meV. Figure 1a illustrates the linear response (black line) and nonlinear absorption spectra measured in transmission geometry of the MoSe₂ monolayer encapsulated in hBN and placed on a diamond substrate.

The exciton transition is excited close to resonance at 1.638 eV with a full width at half maximum (FWHM) of 2.2 meV, -1.7 meV above the center of the exciton resonance, but still well within the 1s exciton absorption line. The 850 fs long pump pulse and the white-light supercontinuum probe pulse are polarized co-linearly with respect to each other. Figure 1a depicts results for a time delay between pump and probe of −90 fs; the field maxima in time of pump and probe overlap on the sample at time zero. Energy densities of 8 μJ/cm² and above invoke a splitting of the exciton resonance in the nonlinear absorption spectrum. An exemplary time evolution of the exciton resonance splitting for an energy density of 16 μJ/cm² is depicted in Fig. 1b on the left. The unperturbed exciton resonance is observed for negative time delays. It splits into two initially fairly symmetric branches as the time delays approach zero. Eventually, these branches re-merge into a single absorption peak for time delays exceeding 0.5 ps. The absorption peak is then broadened and attenuated due to the excitations in the system.

The spectral splitting during temporal overlap is readily explained using a description based on an exciton Bloch equation-of-motion approach[43]. This description accounts for 1s coherent exciton transitions (Eq. (6) in the Supplemental Material (SM)) and incoherent occupations (Eq. (14) in the SM) as well as many-particle Coulomb correlations. The latter account for four-particle Coulomb correlations, i.e. spin-like and spin-unlike biexcitons (Eq. (15) in the SM), and

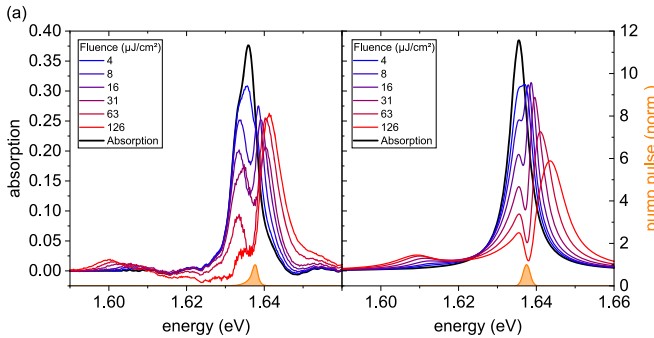
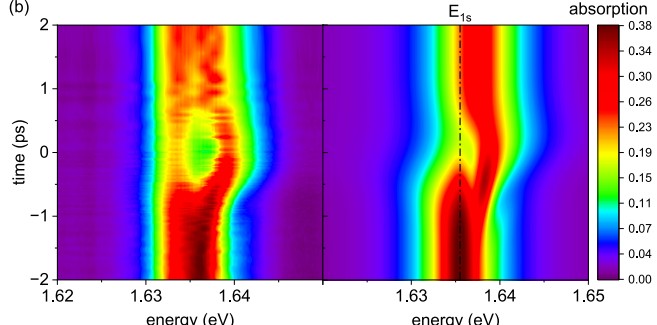

**Fig. 1 | Ultrafast nonlinear absorption dynamics in monolayer MoSe₂: experiment and simulation. a** Nonlinear absorption spectra of the MoSe₂ monolayer sample (left) for co-linear polarization geometries at a nominal pump-probe time delay of −90 fs, shown for various excitation fluences. The linear absorption spectrum (black) and the spectral profile of the optical pump pulse (orange) are

included as references. Corresponding spectra from numerical simulations are displayed on the right. **b** 2D false-color plots of the nonlinear absorption as function of time delay for an excitation energy density of 16 μJ/cm² (left) and the corresponding simulated results (right). $E_{1s}$ marks the 1s exciton energy.

six-particle Coulomb correlations, i.e. spin-like and spin-unlike exciton-biexciton transitions (Eq. (16) and Eq. (17) in the SM), which significantly impact the nonlinear dynamics[16,44–48]. The details are provided in the Supplemental Material. The model takes into account optical field-induced blocking, Coulomb-mediated excitation-induced dephasing, excitation-induced energy shifts and formation of incoherent occupations due to optical interaction as well as exciton-phonon coupling. All excitonic coupling matrix elements involving the many-body effects responsible for the nonlinearities in optically excited semiconductors are calculated microscopically; only the 1s exciton energy, the transition dipole moment and a residual nonradiative broadening unrelated to phonons are adjusted to experimental linear absorption spectra.

Results from the theoretical calculations for the MoSe$_2$ monolayer are shown in the right-hand panels of Fig. 1a, b. In the TMDC case, the exciton binding energy $E_b$ significantly exceeds the Rabi energy $\hbar\Omega = \mathbf{d}^{cv} \cdot \mathbf{E}$, where $\mathbf{d}^{cv}$ represents the transition dipole moment and $\mathbf{E}$ the optical field. At the applied pump powers, the ratio $\hbar\Omega/E_b \sim 10^{-3}$ remains negligible. Additionally, co-linear excitation leads to the formation of spin-unlike biexcitons and exciton-biexciton transitions, as observed in the absorption in Fig. 1 at ~1.60 eV and in the transient differential absorption spectra in Fig. 4. Here, a well-defined peak ~30 meV below the excitonic resonance appears, whose oscillator strength increases with increasing pump fluence. In this regime, the strong Coulomb interaction between quasiparticles in MoSe$_2$ dominates over the light-matter interaction, dictating the observed spectral features and governing the nonlinear response.

Next, we study the scenario where the light-matter interaction is of a similar order of magnitude as the Coulomb interaction of the quasiparticles. The prototypical structure realizing these conditions are (Ga,In)As QWs in co-circular excitation geometry. These have much weaker bound excitons compared to the MoSe$_2$ monolayer, i.e., $\hbar\Omega/E_b \sim 10^{-1}$ at the applied pump fluences and feature no spin-unlike biexcitons and exciton-biexciton correlations. Varying the time delay between the optical pump and probe pulses provides insights into the dynamics of the nonlinear light-matter interaction, as depicted in Fig. 2a. The inset illustrates the linear absorption spectrum of the multiple quantum well (MQW) sample, revealing its narrow 1s exciton absorption peak at 1.467 eV, which is selectively excited by the optical pulse centered at 1.468 eV. At an energy density of 4 $\mu$J/cm$^2$, the exciton resonance remains undisturbed for negative time delays, where the probe pulse arrives before the excitation pulse. However, as the time delay approaches zero, causing temporal overlap between the 1.3 ps-long excitation pulse and the short probe pulse, the exciton

resonance begins to split into two distinct absorption peaks. With increasing temporal overlap, the probe pulse experiences a higher effective driving photon density, leading to more pronounced splitting and the eventual emergence of additional absorption features. This nonlinear response is consistently observed across different excitation pulses of similar duration and spectral range, as long as the exciton is excited near resonance. Notably, significant optical gain emerges between these absorption peaks, even at a moderate energy density of 4 $\mu$J/cm$^2$. This coherent gain feature[49] is evident in Fig. 2a as a deep purple region around 1.469 eV, near the 1s exciton resonance in the linear absorption spectrum. Alongside the spectral Rabi splitting and a coherent gain, the associated Rabi oscillations manifest in the time domain. These oscillations are particularly pronounced for the high-energy branch due to its favorable scattering conditions, which reduce the Coulomb collision rates[24]. The Rabi oscillations can be analyzed quantitatively by examining the time traces spectrally averaged across the gain region, cf. Fig. 2b. A clear single Rabi flop is found at 11 $\mu$J/cm$^2$ while 42 $\mu$J/cm$^2$ yield nearly three full Rabi cycles, well-resolved in theory and experiment.

## Discussion

Next, we explain the key experimental features in Figs. 1 and 2: (a) Rabi oscillations, (b) coherent gain, (c) spectral shifts and splitting and (d) bound biexciton formation for different values of $\hbar\Omega/E_b$ and different excitation geometries, i.e. for dominating Coulomb many-body effects in co-linearly excited TMDCs as well as for moderate Coulomb interaction and comparably more pronounced light-matter coupling in co-circularly excited (Ga,In)As QWs.

(a) *Rabi oscillations*. The microscopic model traces the temporal Rabi oscillations back to oscillations of the total coherently excited 1s exciton density $|P|^2$ and incoherent 1s exciton density $N$.

In the (Ga,In)As MQW, we focus the discussion on $N$, since it provides the strongest contribution. $N$ can be generated by exciton-phonon, exciton-light and exciton-exciton interaction. In the MQW case, incoherent exciton formation via exciton-phonon scattering[50,51] is negligible, cf. Eq. (18). Rabi oscillations are solely determined by Pauli-blocking effects of incoherent excitonic occupations $N$ in fourth order of the optical field, cf. first and second line in Eq. (14) in the SM, since exciton-exciton interaction, cf. third and fourth line in Eq. (14) in the SM, is also of minor importance.

Compared to the MQW, in the monolayer MoSe$_2$, exciton-phonon, Pauli-blocking and exciton-exciton interactions scale differently. Here, in contrast to the (Ga,In)As MQW, optical blocking in the incoherent occupations $N$ is of minor importance, so that this formation

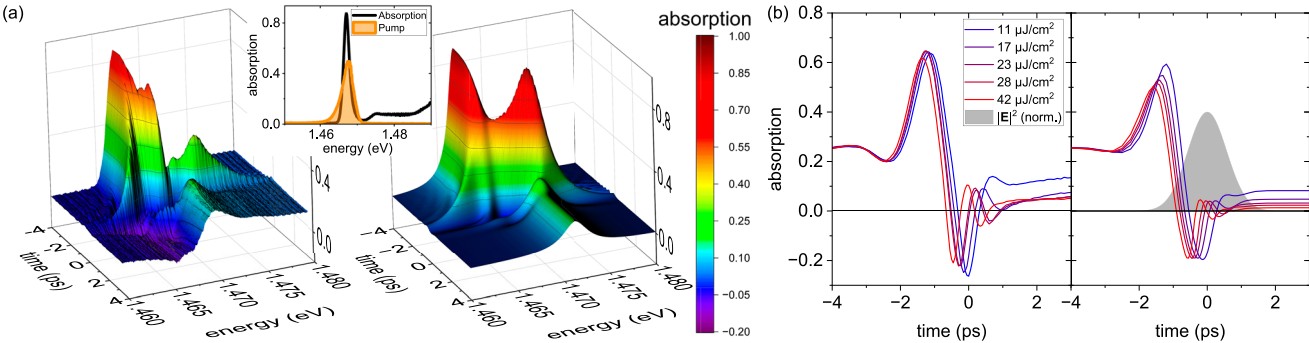

**Fig. 2 | Coherent Rabi oscillations and gain dynamics in (Ga,In)As quantum wells: experiment and microscopic theory. a** Left: 3D false-color representation of the nonlinear absorption for a high-quality (Ga,In)As MQW sample, revealing spectral Rabi splitting and a temporal beat corresponding to Rabi oscillations, particularly prominent in the higher-energy branch. The onset of negative absorption indicates coherent gain. The linear absorption (black) and the pump pulse spectrum (orange) are given in the inset. Right: Microscopically calculated 3D

plot of the nonlinear absorption as a function of time and photon energy at a pump power of 60 $\mu$W (16 $\mu$J/cm$^2$) and a pump detuning of 0.5 meV above the 1s exciton energy for a co-circular pump-probe polarization configuration. **b** Left: Transients through the gain region at 1.469 eV highlighting Rabi oscillations for various optical excitation fluences under a co-circular pump-probe polarization configuration. Right: Corresponding results from microscopic calculations. The grey-shaded area represents the excitation pulse.

mechanism is even outcompeted by exciton-phonon interaction at the applied pump powers: The stronger Coulomb interaction reduces the Pauli-blocking contribution in the incoherent excitonic occupations $N$, Eq. (14) first line, cf. also Tab. SI and SII in the SM, since the excitonic wave functions are more spread out in $\mathbf{q}$-space, and enhances the exciton-exciton interaction in the excitonic transitions $P$, cf. third and fourth line in Eq. (6) in the SM. Thus, the incoherent occupations $N$ do not contribute to the Rabi-flopping dynamics. Similarly, the coherently excited exciton density $|P|^2$ is especially decreased by the excitation-induced dephasing via the biexciton and exciton-biexciton continuum. The different scaling of these mechanisms result as a direct consequence of the stronger confinement in atomically thin TMDC, which increases exciton-exciton and exciton-phonon interaction compared to exciton-light interaction. All in all, in the Coulomb-dominated monolayer $MoSe_2$, no Rabi oscillations are observed.

(b) *Coherent gain*. In the (Ga,In)As QW, the full spectro-temporal dynamics from our calculations, shown in Fig. 2a on the right, also capture the emergence of gain. Its attribution of a coherent nature originates from the two-pulse superposition in the blocking contribution (third term in the first line of Eq. (6) in the SM), which transfers parts of the pump-induced dynamics in the direction of the probe pulse, i.e. it emerges only during the presence of the pump pulse. This "wave-mixing-like" process is significantly different from the more common, incoherent gain due to inversion of an incoherent charge-carrier population. The low excitation density of 16 $\mu J/cm^2$ and the resulting absence of population inversion conclusively rule out any possibility of incoherent gain. Occupation gratings due to Coulomb interaction (second line in Eq. (6) in the SM) and unbound spin-like exciton-biexciton transitions (second term in the last line in Eq. (6) in the SM) play a minor role, but do contribute to an overall enhancement of the coherent gain. In contrast, in the $MoSe_2$ monolayer, the strong Coulomb interaction, coupled with the presence of spin-unlike biexcitons and exciton-biexciton transitions due to co-linear excitation, effectively suppresses the coherent gain signatures at the applied pump powers.

(c) *Energy shifts and splitting*. The narrow linewidth of the MQW sample allows for a detailed quantitative analysis of the light-driven spectral shifts and Rabi splitting using our many-body model. To this end, we examine the individual spectral features induced by the optical excitation. Figure 3a illustrates the measured nonlinear absorption for co-circularly polarized pump and probe pulses at a time delay of 300 fs for increasing photon densities. The exciton resonance begins to split into two distinct absorption peaks at energy densities of 1.4 $\mu J/cm^2$. In particular, the low-energy branch, positioned at 1.4666 eV, maintains its spectral position as the energy density increases, while the high-energy branch shifts from 1.4677 eV to 1.4727 eV, which it reaches at the highest energy density of 99.0 $\mu J/cm^2$. Both branches have narrower linewidths than the linear 1s exciton resonance at low driving photon fluences. Moreover, additional absorption features emerge as the excitation strength increases. A weak resonance appears at lower energies for pump energy densities of 2.8 $\mu J/cm^2$, subsequently shifting from 1.465 eV to 1.460 eV as the excitation is further increased. Another weak absorption peak arises between the two initial branches at 1.468 eV for energy densities of 17.0 $\mu J/cm^2$, maintaining its spectral position even under stronger driving conditions.

Figure 3b traces the spectral positions of all observed absorption peaks relative to the 1s exciton resonance in the linear absorption against the square root of the excitation power, i.e. the field amplitude. In this regime, an approximate analytical formula of the Rabi splitting for vanishing exciton-exciton interaction $V = 0$ and vanishing pump saturation is derived (cf. Eq. (64) in the SM):

$$E_\pm = E_{1s} \pm \sqrt{2D\hbar\Omega_{1s}}, \tag{1}$$

where the Coulomb-enhanced blocking parameter reads: $D = \frac{\sum_\mathbf{q} \varphi_\mathbf{q}^3}{\sum_\mathbf{q} \varphi_\mathbf{q}}$ with the 1s excitonic wave function $\varphi_\mathbf{q}$ at relative momentum $\mathbf{q}$. Eq. (1) is formally equivalent to the splitting observed in a classical 2-LS and therefore shows a linear splitting in the Rabi energy. This linear splitting behavior is observed in the experiment as well as in the full simulations, cf. Fig. 3b. We conclude that the (Ga,In)As MQW sample displays an effective, modified 2-LS response due to its relatively weak Coulomb interaction and the co-circular polarization conditions (no spin-unlike biexcitons or exciton-biexciton transitions). Note that in the simulations, a weak saturation onset at higher fluences can be observed, which is traced back to the onset of optical blocking of the 1s exciton density.

Finally we consider the TMDC monolayer case where the spectral shifts observed in Fig. 1 appear at first glance to be similar to those observed in the QW sample (cf. Fig. 2). The low-energy resonance remains fixed at -1.633 eV, while the high-energy peak continuously shifts to higher energies with increasing excitation, as shown in Fig. 3b

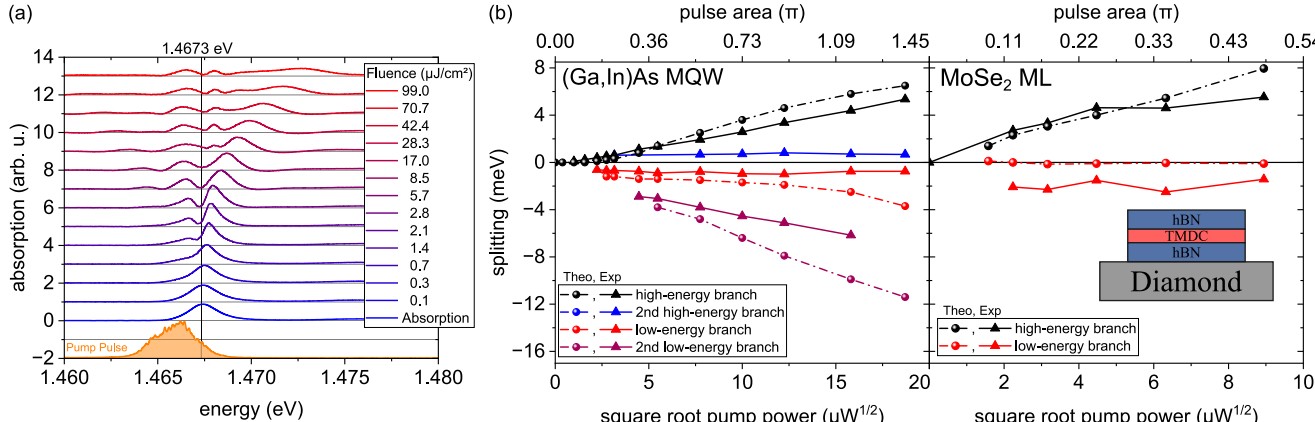

**Fig. 3 | Intensity-dependent Rabi splitting in (Ga,In)As and $MoSe_2$: experiment and microscopic modeling. a** Stacked nonlinear absorption spectra of the (Ga,In) As MQW measured at a 300 fs time delay between co-circular polarized pump and probe pulses for various excitation densities. Rabi splitting is observed above an energy density of 1.4 $\mu J/cm^2$, with additional absorption peaks emerging at higher pump intensities. **b** Energy positions of the absorption peaks from both experiment (triangles) and microscopic modeling (spheres), plotted against the square root of the pump power. Left: (Ga,In)As MQW (10 $\mu W^{1/2} \hat{=} 28.3$ $\mu J/cm^2$). Right: $MoSe_2$ Monolayer (10 $\mu W^{1/2} \hat{=} 157.2$ $\mu J/cm^2$). The inset sketches the sample structure of the $MoSe_2$ monolayer. The pulse area is defined as: Pulse area = $\int_{-\infty}^{\infty} dt\, \Omega^{cv}(t)$ and illustrates the light-matter interaction strength acting directly within the corresponding sample.

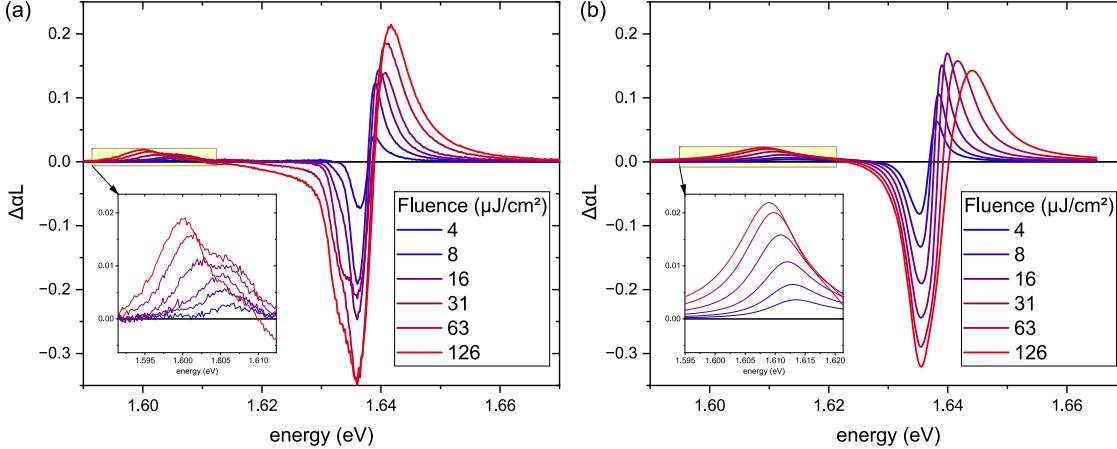

**Fig. 4 | Fluence-dependent biexciton shifts in MoSe₂: experiment vs. theory.** Comparison between experimental (**a**) and theoretical (**b**) $\Delta\alpha L$ spectra of the MoSe₂ ML as a function of excitation fluence, ranging from low (blue) to high fluence (red). A fluence-dependent blueshift of the 1$s$ exciton resonance and a redshift of the excitation-induced biexciton absorption—highlighted in the zoomed-in panel—are observed. These trends are well captured by the microscopic simulations.

on the right, displaying a highly asymmetric splitting. Here, the initial symmetric Rabi splitting into a repulsive (upper) and an attractive (lower) branch is significantly altered by the strong Coulomb interaction, which dominates over the optical interation ($\hbar\Omega/E_b$ -10⁻³): The upper branch exhibits a strong Coulomb-dominated density-dependent blue shift, which is a common behavior of TMDCs[52–54], while the lower branch remains relatively stable as it is more light-dominated (see SM for more details). Notably, a closer analysis reveals a slight sublinear increase of the splitting with the Rabi energy, even though the optically excited exciton density is still far from optical saturation, which deviates from the effective 2-LS behavior, cf. Equation (1). In the MoSe₂ monolayer case, where Coulomb interactions completely overshadow optical interactions, a regime of vanishing Pauli-blocking emerges[55,56] as long as the exciton density is well below the Mott transition[57,58]. Under such conditions, an approximate analytical expression for the Rabi splitting $E_\pm$ at zero detuning can be derived (cf. Eq. (62) in the SM):

$$E_\pm = E_{1s} \pm \sqrt{3}|V|^{\frac{1}{3}}|\hbar\Omega_{1s}|^{\frac{2}{3}}. \tag{2}$$

Here, $E_{1s}$ is the exciton energy, $V$ represents the effective exciton-exciton interaction, and $\hbar\Omega_{1s} = \sum_{\mathbf{q}} \varphi_{\mathbf{q}} \hbar\Omega$ is the excitonic Rabi energy. This Hartree-Fock description already predicts a weak sublinear splitting with respect to the Rabi energy $\hbar\Omega$. The analysis of the microscopic model identifies the origin of this weak sublinear behavior in a regime far from optical saturation: First, we note that the splitting itself originates from an occupation grating rather than a polarization grating, since, at zero pump-probe delay, i.e. maximal splitting, the overall excitonic density is already dominated by incoherent occupations. At that time, four-particle biexcitons (Eq. (15) in the SM) are already decayed and six-particle exciton-biexciton transitions (Eq. (16) and Eq. (17) in the SM) are formed, since they possess a combined coherent (probed transitions) and incoherent (pump-induced excitonic occupation) source. Second, in co-linear excitation, next to spin-like exciton-biexciton transitions, additional spin-unlike exciton-biexciton transitions are induced, which are absent in co-circular excitation. Due to their Coulomb-correlated intervalley nature, they cause a dynamic coupling between both $K$ and $K'$ valleys, which causes an overall attenuation of the Rabi splitting.

The qualitative validity of Eq. (2) results from the appearance of intervalley exciton-biexciton transitions which overcompensate the excitation-induced dephasing by amplifying coherent Coulomb renormalization effects in the probe-induced transition (cf. the second line in Eq. (6) in the SM). The experimentally observed

linewidth narrowing of the splitting peaks corroborates this interpretation: Compared to the 6.7 meV FWHM of the linear absorption resonance, the lower-energy peak narrows to between 3.32 and 6.18 meV, while the higher-energy peak exhibits a FWHM in the range of 3.11 to 4.74 meV as the photon density increases. The simulations also display a linewidth narrowing at small to moderate photon densities.

(d) *Biexciton formation.* An additional resonance on the low-energy side is clearly observed in co-linear measurements in the MoSe₂ monolayer, cf. Figures 1 and 4. Its spectral position is ~30 meV below the excitonic resonance similar to other works, which report biexcitonic binding energies in the range of 20–30 meV[59–62]. Its spectral position experiences a fluence-dependent red shift, which mirrors a density-dependent repulsive interaction between the biexcitonic and the excitonic resonance, as well as an increase in oscillator strength. In co-circular excitation in a (Ga,In)As MQW, no biexcitonic resonance appears. Here, only biexcitonic continua occur, since only one electron-heavy-hole spin configuration is optically addressed. The observed features in the experiments are well reproduced by the microscopic theory, confirming the contributions of bound biexcitons and the biexcitonic continuum.

In summary, we present experimental evidence and theoretical confirmation for Rabi splitting of the 1$s$ exciton resonances in (Ga,In)As QWs and MoSe₂ monolayers under close-to-resonant excitation conditions. The fundamentally different nature of both 2-LS, particularly regarding the interplay of Coulomb interactions and light-matter coupling, in conjunction with specific excitation conditions, dictates the splitting behavior, occurrence of Rabi oscillations, and coherent gain: In (Ga,In)As MQWs with co-circular excitation and weaker Coulomb interaction, the Rabi splitting scales nearly linearly with the Rabi energy and is accompanied by pronounced Rabi oscillations and coherent optical gain. Conversely, in MoSe₂ monolayers, with co-linear excitation and stronger Coulomb interaction, only sublinear Rabi splitting is observed, with no evidence of Rabi oscillations or gain. Our findings are corroborated by a microscopic theory that elucidates the physical mechanisms underlying these effects. These insights are pivotal for leveraging exciton-based coherent phenomena in next-generation ultrafast optoelectronic and switching devices.

## Methods
### Sample fabrication
We investigate two distinct types of samples. The first sample is a MoSe₂ monolayer (ML) encapsulated in hBN and placed on a diamond substrate. This configuration promotes high optical quality, a narrow

exciton linewidth, and improved environmental stability due to the protective hBN encapsulation.

The second sample is a type-I band alignment multi-quantum well structure. High-resolution X-ray diffraction (HRXRD) measurements reveal that the $Ga_{0.942}In_{0.058}As$ QW layers are 7.6 nm thick. These QWs are arranged in a stack of 10 layers, separated by barriers composed of GaAs/GaAsP/GaAs, which are designed to provide strain compensation. The barrier thickness is 28.6 nm. Atomic force microscopy (AFM) measurements reveals a well-defined surface quality with a root-mean-square roughness of 0.3 nm. Additional details on the growth process and structural characterization are provided in Ref. [63].

Both samples exhibit a pronounced and well-defined 1s exciton resonance in their linear absorption spectra, which can be resonantly excited using an optical pulse.

### Experimental details

For the investigation of the MQW sample, we utilize a regenerative amplifier system operating at a 5 kHz repetition rate. This system delivers 50 fs pulses centered around 800 nm with a pulse energy of 1.6 mJ. A small portion of the amplifier output generates a white-light supercontinuum in a 4 mm thick sapphire-crystal, while the remaining output drives an optical parametric amplifier (OPA). The OPA provides tunable pulses at the desired central wavelength, which are then spectrally narrowed using a pulse shaper. After shaping, the pulses have a duration of 1345 fs. The shaped pulses are then focused onto the sample to a spot size of 300 $\mu$m. In the white-light beam path, a wedge beamsplitter is used to split the light into two beams: one is focused down to a 200 $\mu$m spot on the sample to probe the excitation-induced absorption changes, while the other serves as a reference pulse. The sample is held at liquid helium temperatures in a cold-finger cryostat. After passing through the sample, the white-light supercontinuum and the reference pulse are spectrally analyzed using an imaging spectrometer equipped with a 600 lines/mm grating and an sCMOS camera (see Fig. 5). The sCMOS detector has 2160 lines that can be read out individually, allowing multiple lines to be grouped into a region of interest (ROI). By imaging the white-light pulse transmitted through the sample and the reference pulse onto different lines of the detector, both spectra can be captured simultaneously and independently. This setup allows for the calculation of a transfer function ($T_f$) that converts the reference spectrum ($T_{ref}$) into the spectrum of the pulse transmitted through the unexcited sample. The reference path thus provides the spectrum of the unexcited sample at any time, compensating for fluctuations in the white-light spectrum, as the transfer function remains stable over time. By introducing optical excitation, we can simultaneously measure the transmission through the excited sample ($T_P$) and the transmission through the unexcited sample ($T_0$). At the beginning of each measurement, the photoluminescence background ($T_{Pl}$) and the scattered light background ($T_{Bg}$) in both paths are recorded. The differential absorption at each time step is then calculated as follows:

$$\Delta\alpha L = -\ln\left(\left(T_P - T_{Pl}\right)/\left(T_f \cdot \left(T_{ref} - T_{Bg}\right)\right)\right). \tag{3}$$

This method offers several advantages: the simultaneous measurement of the excited and unexcited sample transmissions minimizes distortion from white-light fluctuations, and each time step can be monitored in real time by simply reading out the spectrometer, resulting in a highly accurate, fast, and low-noise method to determine differential absorption. Additionally, this approach can also be used to determine the linear absorption of the sample. To do this, we first remove the sample from the beam path to establish the transfer function that converts the reference path into the spectrum of the pulse transmitted through the sample holder. The sample is then reintroduced, allowing us to measure the transmission through the unexcited sample and calculate the linear absorption. Adding the linear absorption to the differential absorption then yields the total absorption of the excited sample.

To measure the $MoSe_2$ monolayer (see Fig. 6 for a brightfield image), we utilized an amplifier laser system operating at a 100 kHz repetition rate with a central wavelength of 1030 nm. As before, a portion of the amplifier output is used to generate a white-light supercontinuum in a 4 mm thick sapphire-crystal, while the remaining output drives an OPA. The output from the OPA is spectrally narrowed using a combination of short-pass and long-pass filters. The shaped pulse is then focused onto the sample to a spot size of 28.5 $\mu$m to excite the sample. The probing white-light supercontinuum is expanded to a diameter of 1 inch before being focused down to an ~5 $\mu$m spot on the sample using a 1-inch lens with a 4 cm focal length. The transmitted light is then focused into a spectrometer for spectral analysis. To monitor the spot position on the sample, the setup allows for imaging the transmitted light onto a camera. By strongly attenuating the white-light probe and using additional background illumination, the probe position on the monolayer sample can be precisely observed.

For the $MoSe_2$ monolayer measurements, we employ a systematic approach using mechanical shutters in both the optical excitation and white-light probing paths to capture various spectra at each time step. This method allows us to record the light transmitted through the unexcited sample ($T_0$), the spectrum of the excited sample ($T_P$), a background measurement ($T_{Bg}$), and the photoluminescence background ($T_{Pl}$). These measurements enable us to calculate differential absorption via:

$$\Delta\alpha L = -\ln\left(\left(T_P - T_{Pl}\right)/\left(T_0 - T_{Bg}\right)\right). \tag{4}$$

Since the diamond substrate introduces light scattering that affects transmission through the sample, directly measuring the linear absorption by comparing the light transmitted through the sample with that through the sample holder alone is not feasible. To address this challenge, we analyze the transmitted light through the sample itself. The resulting spectrum shows a distinct dip corresponding to the exciton resonance of the monolayer, as illustrated as gray line in Fig. 7. Due to the high exciton binding energy in the $MoSe_2$ monolayer, the spectrum is dominated by the exciton resonance, which is further corroborated by the differential absorption measurements. To create a reference spectrum, we fit a polynomial function to the transmitted light spectrum, deliberately excluding the exciton dip. The polynomial

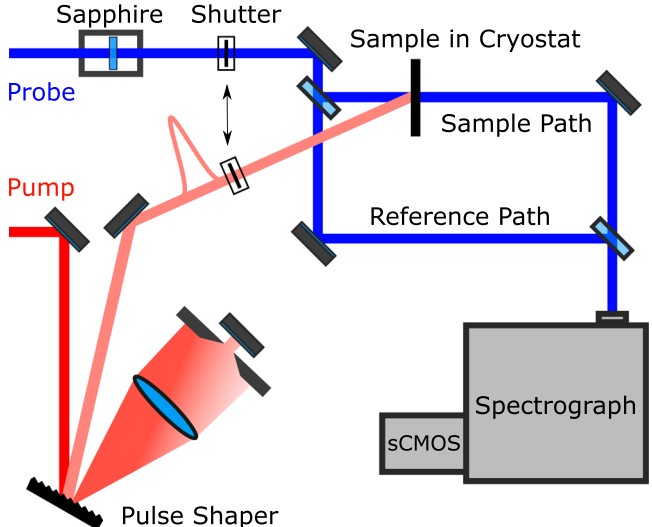

**Fig. 5 | Schematic of the optical pump-probe setup with pulse shaper and reference path.** Illustration of the optical pump−optical probe setup, featuring a pulse shaper to tailor the excitation pulse and two probe paths: one directed through the sample and the other serving as a reference.

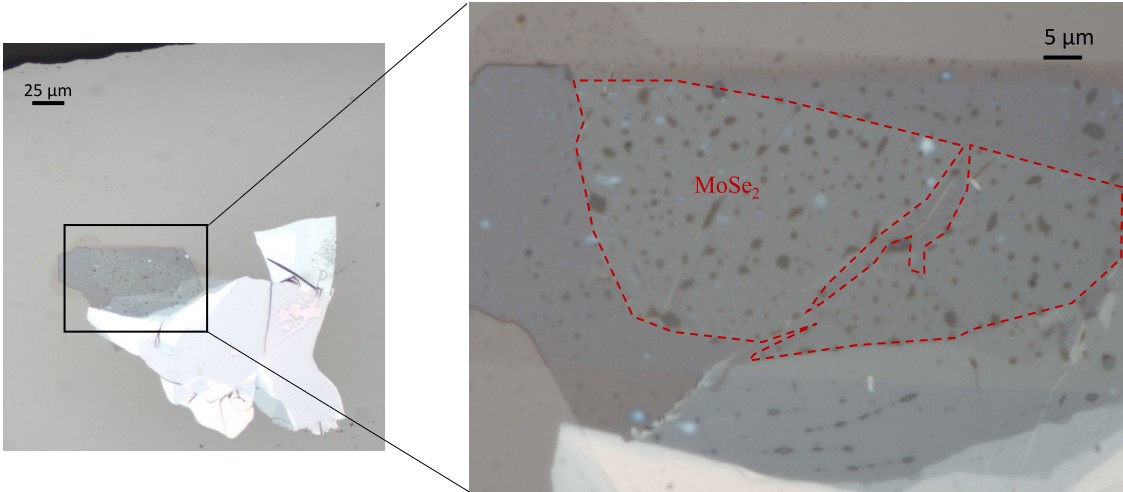

**Fig. 6 | Brightfield microscopy of the TMDC sample with magnified view of the MoSe₂ monolayer area.** Brightfield image of the TMDC sample. The right side displays an enlarged view of the monolayer MoSe₂, marked with a red outline to indicate the specific region where measurements were taken.

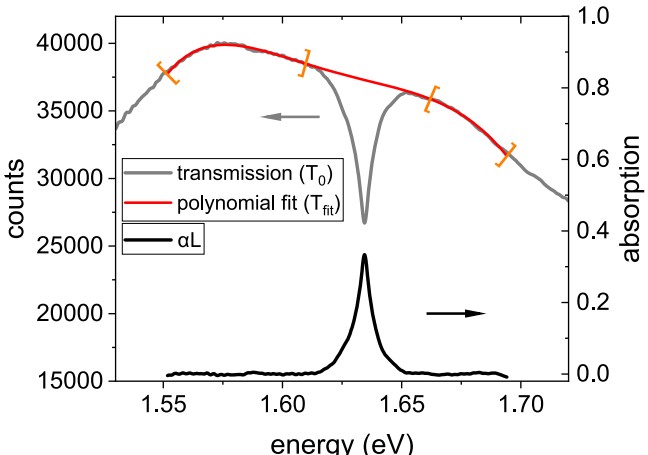

**Fig. 7 | Linear absorption spectrum of monolayer MoSe₂ derived from the transmission spectrum.** Transmission spectrum of the MoSe₂ monolayer. The transmitted light (gray) shows a dip at the exciton resonance. The polynomial fit (red) excludes this dip and serves as the reference spectrum for calculating linear absorption. The fit boundaries are marked in orange, and the calculated linear absorption is shown in black.

fit and its boundaries are shown in Fig. 7. This polynomial fit serves as the reference white-light spectrum, which we use to calculate the sample's linear absorption via

$$\alpha L = -\ln\left(\frac{T_0}{T_{\text{fit}}}\right). \tag{5}$$

This approach provides the most accurate estimate of the sample's linear absorption around the exciton resonance. By adding the differential absorption to the linear absorption, we obtain the nonlinear absorption of the sample at each time step.

## Theory
We account for optical light-matter and electron-electron, hole-hole and electron-hole Coulomb interaction[64] as well as Coulomb exchange interaction[65,66] in second quantization, apply the unit operator method[43,67], perform a cluster expansion[68] with respect to electron-hole

pair operators and formulate the theory in the emerging correlated expectation values. The direct Coulomb interaction and optical interaction are treated up to fourth order within the dynamics-controlled truncation scheme (DCT)[69], which is necessary to correctly capture the optical formation of incoherent excitonic occupations. This is especially important in quantum wells at cryogenic temperatures, where the formation of incoherent excitonic occupations due to exciton-phonon interaction is negligible. Coulomb exchange interaction is included only up to third order DCT, since higher orders are expected to be negligible. Moreover, we include the formation of incoherent excitonic occupations via exciton-phonon scattering[50,51] on a second-order DCT level[70]. Density-dependent exciton-phonon interaction is neglected[71,72]. Since we are mainly interested in the dynamics within and shortly after the duration of the optical pump pulse, the optical formation of excitonic occupations is much more important than a redistribution of momenta due to Coulomb scattering[73,74]. Therefore, we do not consider fully incoherent excitonic Coulomb scattering.

All in all, we consider correlated expectation values for up to six particles[68], i.e., we account for one-, two- and three-exciton correlations: The excitonic transitions $P = \langle v^\dagger c\rangle_c$, the excitonic occupations $N = \langle c^\dagger v v^\dagger c\rangle_c$, the biexcitonic correlations $B = \langle v^\dagger c v^\dagger c\rangle_c$ and the exciton-biexciton correlations $R = \langle c^\dagger v v^\dagger c v^\dagger c\rangle_c$. Here $v/c^{(\dagger)}$ denote the valence/conduction band annihilation (creation) operators. All many-body correlations are first established via the Heisenberg equations of motion in the electron-hole picture and then transformed into the excitonic picture by an expansion in the solutions of the one-exciton Schrödinger equation (Wannier equation)[43]. The biexcitons $B$ and exciton-biexciton transitions $R$ are further expanded in solutions of the two-exciton Schrödinger equation[16,44]. The subscript "$c$" denotes the correlated part of the corresponding expectation value, where all lower order correlations have been removed. We consider only $1s$ excitonic transitions $P$ and occupations $N$, but include $1s$, $2s$ and $3s$ states in the calculation of the two-exciton states.

Since the purpose of the theory is a qualitative understanding of the experiments, we make the assumption $N_Q \approx N\delta_{Q,0}$, which results in an effective momentum-independent formulation of the incoherent dynamics. A full momentum-dependent treatment of the incoherent exciton dynamics, which complicates the numerical evaluation of the excitonic occupations $N$ and especially the exciton-biexciton correlations $R$ due to their partly incoherent nature, is beyond the scope of this work. A detailed description of the theory and the calculations can be found in the SM.

## Data availability

The datasets generated in this study have been deposited in the JLU-pub database https://doi.org/10.22029/jlupub-20064.

## Code availability

Code pertaining to the theoretical model in this work will be made available upon request to the corresponding author.

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

## Acknowledgements

S.C., F.S., M.S., C.F. and K.V. acknowledge financial support from the Deutsche Forschungsgemeinschaft (DFG) via the Collaborative Research Center SFB 1083 (Project No. 223848855). H.M. and A.K. acknowledge financial support by the DFG through Project KN 427/11-2, Project No. 420760124. A.K. acknowledges financial support by the DFG through Project KN 427/15-1, Project No. 556436549. A.C. and M.C. acknowledge financial support by the DFG through Würzburg-Dresden Cluster of Excellence on Complexity and Topology in Quantum Matter ct.qmat (EXC 2147, Project-ID 390858490) and SFB 1277 (project B05, Project-ID: 314695032). K.W. and T.T. acknowledge support from the JSPS KAKENHI (Grant No. 21H05233 and 23H02052) and World Premier International Research Center Initiative (WPI), MEXT, Japan.

## Author contributions

A.K. and S.C. conceived the study. C.F., and K.V. designed and epitaxially grew the quantum well sample. M.C. and A.C. exfoliated the monolayer sample, encapsulated it in h-BN and placed it on a diamond substrate. K.W. and T.T. provided the hBN. F.S., M.S., D.A., I.M., F.D. and S.C. set up and carried out the experiments and analysed the experimental data. H.M., O.V., L.G., and A.K. developed the microscopic theory and carried out the numerical calculations. F.S., H.M., M.S., A.K. and S.C. wrote the manuscript with contributions from all authors.

## Funding

## Competing interests

The authors declare no competing interests.
