## [Transparent Peer Review file · Nature Communications]

Rabi Splitting in Confined Semiconductors: MoSe₂ Monolayers vs. (Ga,In)As Quantum Wells

Corresponding Author: Dr Markus Stein

Version 0:

Reviewer comments:

Reviewer #1

(Remarks to the Author)

The authors investigated the Rabi splitting dynamics in two paradigmatic 2D solid-state systems: a hBN-encapsulated MoSe₂ monolayer and a (Ga,In)As multiple quantum well structure. Features beyond Rabi splitting such as asymmetric Rabi oscillations, the emergence of additional spectral resonances, and coherent gain are captured by a microscopic theory based on Heisenberg equations of motion and an exciton expansion.

However, the manuscript should be improved a lot before the reconsideration of publication in Nature Communications.

1. I think the topic of this manuscript is narrow and not of general new interest. Rabi splitting of similar works has already been reported for a large variety of systems. The general physics seems to be understood in the literature, (such as in recent Nanophotonics 12(16): 3267–3275(2023) or previous Nature Physics volume 2, pages81–90 (2006)) and it remains unclear to me where the new physics is hidden in this new manuscript.

2. Looking at Fig. 5, I can see a lot of defects on-top of the WS₂ sample. The authors should clarify the influence of so many bubbles or defects.

3. Then the experimental part is rather disappointing as the authors only looked into a single pumping frequency and could not even verify the whole loop of Rabi oscillations within several pico-seconds.

4. I cannot find the reason of comparing the Rabi splitting of MoSe₂ monolayers vs. (Ga,In)As quantum wells, which are two different systems. The authors should elaborate more about the topic with data/figure and text.

5. Moreover, the SI contains much important things than the main text. I think the authors would be well advised to exchange the experimental/setup figures for some theory figures from the SI.

6. The title and the abstract is not appealing at all. The paper does not even marginally discuss heat conductivity of the system. How can the findings be used on the order of the Rabi oscillation period? All these promising discussions seem to be completely missing.

Reviewer #2

(Remarks to the Author)

This manuscript presents a compelling comparative study of Rabi splitting dynamics in two paradigmatic 2D systems—MoSe₂ monolayers and (Ga,In)As quantum wells. The work combines rigorous experiments with a sophisticated microscopic many-body theory, offering novel insights into the interplay between Coulomb interactions and light-matter coupling. The findings are of broad interest to the fields of quantum optics, condensed matter physics, and nanophotonics. This work makes a significant contribution to understanding light-matter interactions in 2D systems. Nonetheless, the following points should be addressed in full:

1. Why the splitting of MoSe₂ monolayer is asymmetric? Please provide more detailed discussions.
2. The low-energy branch in the MoSe₂ monolayer simulation is almost missing, which shows a clear deviation with the experimental results.
3. What are the changes of MoSe₂ monolayer time evolution of the exciton resonance splitting under different excitation

energy densities, and whether this splitting is adjustable? I suggest that the authors should provide more information about this.

4. Why do (Ga, In) As quantum wells use 3D false-color representation, which is different from 2D false-color plots of MoSe₂ monolayer? I think it may be more intuitive to use the same representation method for comparison.

5. Can the Rabi oscillations be observed in the transient spectra of MoSe₂ monolayer, like those of (Ga, In) As quantum wells?

6. Although the authors have made a detailed theoretical explanation for the Rabi oscillations in the experimental results of (Ga, In) As quantum wells, is there any way to rule out the possibility of phonon oscillations?

7. Why does the authors use such a wide pulse width (1345fs) laser as a pump source for measurement?

8. I wonder why the authors use two different lasers for measuring, as it seems inappropriate to compare the results obtained under different experimental conditions.

Version 1:

Reviewer comments:

Reviewer #1

(Remarks to the Author)

The authors have addressed most of my concerns, thus I support the publication in NC.

Reviewer #2

(Remarks to the Author)

I thank the authors for their response. After thorough revisions, the authors have adequately addressed my concerns. I recommend publication in Nature Communications.

Detailed Responses to the Referee Remarks Report of the First Referee:

The authors investigated the Rabi splitting dynamics in two paradigmatic 2D solid-state systems: a hBN-encapsulated MoSe₂ monolayer and a (Ga,In)As multiple quantum well structure. Features beyond Rabi splitting such as asymmetric Rabi oscillations, the emergence of additional spectral resonances, and coherent gain are captured by a microscopic theory based on Heisenberg equations of motion and an exciton expansion. However, the manuscript should be improved a lot before the re-consideration of publication in Nature Communications.

Our Response:

We thank the referee for carefully assessing our manuscript and for providing constructive comments. Having been granted the opportunity by the editor to revise our manuscript for Nature Communications, we have addressed each of the comments below in detail.

Referee 1:

1. I think the topic of this manuscript is narrow and not of general new interest. Rabi splitting of similar works has already been reported for a large variety of systems. The general physics seems to be understood in the literature, (such as in recent Nanophotonics 12(16): 3267–3275(2023) or previous Nature Physics volume 2, pages81–90 (2006)) and it remains unclear to me where the new physics is hidden in this new manuscript.

Our Response:

We thank the referee for raising this point and definitely respect the referee’s viewpoint. In particular, the cited recent works demonstrate the timeliness of our work and the ongoing interest in effects related to Rabi splitting. We kindly want to iterate that yet our study offers three distinct advances:

i) Cavity-free Rabi splitting in a TMDC monolayer.

Our work demonstrates clear Rabi splitting in a stand-alone, i.e. cavity-less hBN-encapsulated MoSe₂ monolayer. Unlike previous studies that require microcavities or plasmonic resonators to enhance light–matter interaction in TMDCs, our work is the first unambiguous observation of such strong coupling without any engineered cavity, This offering a novel route toward much more straight forward, ”intrinsically” polaritonic systems.

ii) First head-to-head comparison of two Coulomb regimes.

This side-by-side study directly compares Rabi splitting in a “pure” 2D excitonic system (MoSe₂ monolayer, where Coulomb interactions dominate) versus a conventional III–V quantum well (where light–matter coupling and Coulomb binding are of comparable magnitude): This approach to clearly different interaction regimes under comparable experimental conditions in one general theoretical description allows us to elucidate how the strength of the Coulomb interaction quantitatively modifies the Rabi splitting and identify higher-order contributions and their manifestation in optical spectra.

iii) Quantitative insights from a microscopic many-body theory.

Our theoretical approach, based on a microscopic many-body framework using Heisenberg equations of motion, goes significantly beyond the standard phenomenological two-oscillator model. It captures the distinct scaling of the Rabi splitting with exciton binding energy in each material system as well as providing a consistent explanation for emergent features observed in the experiment, such as asymmetric spectral line shapes, additional resonance peaks, and coherent gain. These effects are rooted in many-body interactions and are not accessible within more approximative and phenomenological models, highlighting the predictive power and necessity of our microscopic treatment. In particular, the interplay of excitons, biexcitons and exciton-biexciton transitions is included.

All in all, our study uncovers truly new physics by (i) demonstrating strong coupling without any cavity in a TMDC monolayer, (ii) presenting the first controlled comparison between two markedly different Coulomb regimes, and (iii) providing a unified microscopic theory that quantitatively accounts

for the observed differences. We hope that this clarifies the broad significance and originality of our manuscript going beyond the current state-of-the-art research around the fundamental phenomenon of Rabi splitting.

Referee 1:

2. Looking at Fig. 5, I can see a lot defects on-top of the WS₂ sample. The authors should clarify the influence of so many bubbles or defects.

Our Response:

The preparation of the samples involves encapsulation of the MoSe₂ monolayers between thin sheets of hBN placed on a diamond substrate. The hBN is required to suppress disorder [1] by providing smooth interfaces and homogeneous dielectric environment. As a consequence of encapsulation and subsequent annealing of the samples in high-vacuum, the adsorbants trapped at the interfaces aggregate in larger bubbles. It occurs as the consequence of the van der Waals forces between the layers pushing the impurities out, while leaving behind clean interfaces in close contact on atomic scale [2]. This process of self-cleaning is well-known [3, 4] and typically requires preliminary mapping of sample properties prior to subsequent measurements to identify areas with good monolayer-hBN contact.

In our study we performed hyperspatial photoluminescence mapping of the sample using low-power continuous-wave excitation at liquid helium temperatures. The extracted energy of the neutral exciton for the studied MoSe₂ monolayer is presented in Fig. 1 (a). The map shows regions with strong fluctuations of the exciton energy due to both substrate-induced strain and formation of bubbles. Most importantly, it allows us to identify sufficiently large areas on a scale of 10 μm with smooth potential profiles. These areas feature characteristic optical features of high-quality MoSe₂ with narrow spectral lines due to suppressed disorder, see example in Fig. 1 (a). All subsequent measurements using pump-probe spectroscopy are performed on such areas with good interlayer contact in absence of bubbles and strain fluctuations.

The figure is now included in the SI to demonstrate the procedure of identifying suitable regions on the sample by hyperspatial mapping.

Figure 1: (a) Spatially-resolved resonance energy of the neutral exciton resonance (X_0) extracted from hyperspatial photoluminescence mapping. A continuous-wave laser with 532 nm wavelength was used for excitation with a low power density of 90 W/cm² focused on a 1 μm spot. The step size for scanning was set to 2 μm and the measurements were performed at the heat-sink temperature of 5 K. (b) Characteristic PL spectrum obtained in a region with smooth energy profile, as indicated in panel (a). The spectrum shows characteristic features of neutral and charged excitons with narrow line-widths as expected for high-quality hBN-encapsulated samples.

Referee 1:

3. Then the experimental part is rather disappointing as the authors only looked into a single pumping frequency and could not even verify the whole loop if rabi oscillations within several pico-seconds.

Our Response:

We regret that the referee considers the experimental part disappointing, but we would like to take this opportunity to clarify our experimental approach. The manuscript features only prototypical results obtained at representative excitation energies to clearly discuss the physics and best disentangle the individual observations by our microscopy theory.

We want to emphasize that we have tested a wide range of excitation pulses covering many different excitation frequencies as well as different spectral widths for all samples. We have added a representative selection of excitation-energy-dependent measurements for the MoSe₂ sample in the revised Supplementary (c.f. Figure 9) to explicitly clarify this point. These data demonstrate the robustness of the observed spectral Rabi splitting across multiple detunings. Out of the large manifold of data, we selected the excitation conditions that provided the clearest and most symmetric Rabi splitting, as also confirmed by theory resulting in the exemplary results presented in the main manuscript.

Next, we want to address the issue raised regarding the “full loop” of temporal Rabi oscillations: We have indeed observed clear and coherent Rabi oscillations over multiple cycles in the (Ga,In)As MQW system, as shown and discussed in the manuscript. In contrast, we cannot observe temporal Rabi oscillations in the MoSe₂ monolayer due to physical reasons rather than for experimental limitations, such as the signal-to-noise ratio. The absence of temporal Rabi oscillations is instead a direct consequence of the fundamental many-body interactions within the monolayer system. Our microscopic theory attributes this in MoSe₂ to two key mechanisms: (i) Enhanced exciton-exciton interaction due to enhanced Coulomb interaction via reduced screening, and (ii) enhanced exciton-phonon interaction due to the enhanced confinement compared to a (Ga,In)As MQW. Both mechanisms result in reduced Pauli-blocking dynamics, which are essential in the occurrence of Rabi flops, in the incoherent exciton density N , so that Rabi oscillations in the Coulomb-dominated monolayer MoSe₂ are suppressed.

We have updated the corresponding discussion in the manuscript, which now includes the following:

(a) Rabi oscillations. The microscopic model traces the temporal Rabi oscillations back to oscillations of the total coherently excited 1s exciton density $|P|^2$ and incoherent 1s exciton density N .

In the (Ga,In)As MQW, we focus the discussion on N , since it provides the strongest contribution. N can be generated by exciton-phonon, exciton-light and exciton-exciton interaction. In the MQW case, incoherent exciton formation via exciton-phonon scattering is negligible, cf. Eq. (17). Rabi oscillations are solely determined by Pauli-blocking effects of incoherent excitonic occupations N in fourth order of the optical field, cf. first and second line in Eq. (13) in the SM, since exciton-exciton interaction, cf. third and fourth line in Eq. (13) in the SM, is also of minor importance.

Compared to the MQW, in the monolayer MoSe₂, exciton-phonon, Pauli-blocking and exciton-exciton interactions scale differently. Here, in contrast to the (Ga,In)As MQW, optical blocking in the incoherent occupations N is of minor importance, so that this formation mechanism is even outcompeted by exciton-phonon interaction at the applied pump powers: The stronger Coulomb interaction reduces the Pauli-blocking contribution in the incoherent excitonic occupations N , Eq. (10) first line, cf. also Tab. SI and SII in the SM, since the excitonic wave functions are more spread out in \mathbf{q} -space, and enhances the exciton-exciton interaction in the excitonic transitions P , cf. third and fourth line in Eq. (6) in the SM. Thus, the incoherent occupations N do not contribute to the Rabi-flopping dynamics. Similarly, the coherently excited exciton density $|P|^2$ is especially decreased by the excitation-induced dephasing via the biexciton and exciton-biexciton continuum. The different scaling of these mechanisms result as a direct consequence of the stronger confinement in atomically thin TMDC, which increases exciton-exciton and exciton-phonon interaction compared to exciton-light interaction. All in all, in the Coulomb-dominated monolayer MoSe₂, no Rabi oscillations are observed.

Therefore, we hope this extended discussion clarifies that the absence of temporal Rabi oscillations

in MoSe₂ is due to physical reasons explained by our microscopic model rather than a limitation of the experimental design. We have revised the manuscript and supplementary material accordingly to make these points more explicit.

Referee 1:

4. I cannot find the reason of comparing the rabi splitting of MoSe₂ monolayers vs. (Ga,In)As quantum wells, which are two different systems. The authors should elaborate more about the topic with data/figure and text.

Our Response:

We thank the referee for pointing out that this important motivation was not sufficiently emphasized in the manuscript. Indeed, MoSe₂ monolayers and (Ga,In)As quantum wells are fundamentally different material systems, with distinct dimensionalities, dielectric environments, and excitonic properties. Precisely these differences have motivated our side-by-side comparison to elucidate the physical consequences. Our study aims to highlight how the balance between Coulomb interaction and light-matter coupling shapes the Rabi splitting dynamics across two prototypical and contrasting 2D semiconductor systems rather than equating these two material systems. The MoSe₂ monolayer serves as a prototypical example material system where Coulomb interactions are strong and dominate over the light-matter coupling, while the (Ga,In)As quantum well represents a system where the light-matter interaction is comparable to, or even stronger than, the Coulomb interaction in the material. This contrast results in qualitatively different physical phenomena: Rabi oscillations, coherent gain, and additional spectral sidebands clearly emerge in the quantum well system. All these features are strongly suppressed in the monolayer due to the strong many-body Coulomb effects, including biexciton formation and enhanced dephasing. Interpreting both systems within a unifying microscopic many-body theory enables us to unambiguously clarify the effect of Coulomb interaction strength on the Rabi dynamics in a controlled, quantitative manner. We believe that this comparative analysis provides novel and broadly relevant insights into the non-perturbative regime of light-matter interaction in semiconductors. To better communicate this motivation and its significance, we have revised the Introduction and Discussion sections of the manuscript and now more explicitly articulate the rationale behind our comparative approach, as well as the novel insights that arise from it.

Referee 1:

5. Moreover, the SI contains much important things than the main text. I think the authors would be well advised to exchange the experimental/setup figures for some theory figures from the SI.

Our Response:

We appreciate the referee's helpful suggestion and follow his suggestion to integrating more theoretical content into the main manuscript to strengthen the scientific narrative. In response to the referee's recommendation, we have revised the manuscript layout to bring several of the key theoretical results from the Supplementary Information into the main text. This restructuring enhances the balance between experiment and theory and better highlights the predictive power and physical insight provided by our microscopic many-body approach.

Referee 1:

6. The title and the abstract is not appealing at all. The paper does not even marginally discuss heat conductivity of the system. How can the findings be used on the order of the rabi oscillation period? All these promising discussion seem to be completely missing.

Our Response:

We thank the referee for this constructive feedback and regret that the original title and abstract did not adequately reflect the focus and novelty of our work. We have thus revised the abstract to more clearly highlight the core contribution of the manuscript: the first comparative demonstration of cavity-free Rabi splitting dynamics in two prototypical 2D semiconductors (hBN-encapsulated MoSe₂ monolayers and (Ga,In)As quantum wells) and its unified microscopic many-body explanation. This explanation reveals how the balance between Coulomb interactions and light-matter coupling governs the scaling of the splitting, the emergence (or absence) of Rabi oscillations, and coherent gain.

We also appreciate the opportunity to clarify the role of heat conductivity: Our study focuses exclusively on the regime of ultrafast coherent light–matter interaction. These phenomena occur on femtosecond to sub-picosecond timescales and are much faster than those relevant for thermal effects such as lattice heating, phonon-mediated relaxation, or heat diffusion, which typically evolve over tens to hundreds of picoseconds [5]. Regardless, we have explicitly included exciton-phonon interactions in our microscopic Heisenberg-equation-based theory according to Ref. [6] to verify the negligible role of thermal processes:

$$\begin{aligned}\partial_t P|_{\text{X-phonon}} &= -\gamma_{\text{phon}} P, \\ \partial_t N_{\mathbf{Q}}|_{\text{X-phonon}} &= \Gamma_{\mathbf{Q}}^{\text{form}} |P|^2 + \sum_{\mathbf{K}} \Gamma_{\mathbf{K},\mathbf{Q}}^{\text{in}} N_{\mathbf{K}} - \sum_{\mathbf{K}} \Gamma_{\mathbf{K},\mathbf{Q}}^{\text{out}} N_{\mathbf{Q}}.\end{aligned}\quad (1)$$

The outscattering rate reads:

$$\begin{aligned}\Gamma_{\mathbf{K},\mathbf{Q}}^{\text{out}} &= \frac{2\pi}{\hbar} \sum_{\pm,\alpha,q_z} |G_{\mathbf{Q}-\mathbf{K},q_z,\alpha}^e - G_{-\mathbf{Q}+\mathbf{K},q_z,\alpha}^h|^2 \left(\frac{1}{2} \pm \frac{1}{2} + n_{\pm\mathbf{Q}\mp\mathbf{K},q_z,\alpha} \right) \\ &\quad \times \delta(E_{\mathbf{K}} - E_{\mathbf{Q}} \pm \hbar\omega_{\pm\mathbf{Q}\mp\mathbf{K},q_z,\alpha}),\end{aligned}\quad (2)$$

where \mathbf{Q}, \mathbf{K} are the two-dimensional excitonic center-of-mass momenta, q_z is the out-of-plane phonon momentum, α is the phonon mode, $G_{\mathbf{K},q_z,\alpha}^{e/h}$ are the exciton-phonon interaction matrix elements, $n_{\mathbf{K},q_z,\alpha}$ is the phonon occupation (Bose-Einstein distribution), $E_{\mathbf{Q}} = E + \frac{\hbar^2 \mathbf{Q}^2}{m_e + m_h}$ is the exciton dispersion and $\hbar\omega_{\mathbf{K},q_z,\alpha}$ is the phonon dispersion. The inscattering rate is obtained by:

$$\Gamma_{\mathbf{K},\mathbf{Q}}^{\text{in}} = \Gamma_{\mathbf{Q},\mathbf{K}}^{\text{out}}.\quad (3)$$

The exciton-phonon interaction matrix elements read:

$$G_{\mathbf{K},q_z,\alpha}^e = \sum_{\mathbf{q}} \varphi_{\mathbf{q}+\beta\mathbf{K}}^* \varphi_{\mathbf{q}} g_{\mathbf{K},q_z,\alpha}^c F_{q_z}, \quad G_{\mathbf{K},q_z,\alpha}^h = \sum_{\mathbf{q}} \varphi_{\mathbf{q}-\alpha\mathbf{K}}^* \varphi_{\mathbf{q}} g_{\mathbf{K},q_z,\alpha}^v F_{q_z},\quad (4)$$

where $\varphi_{\mathbf{q}}$ are the excitonic wave functions and F_{q_z} the confinement form factors [6]. The electron-phonon interaction potentials $g_{\mathbf{K},q_z,\alpha}^{c/v}$ are given by:

$$g_{\mathbf{K},q_z,\alpha}^{c/v} = \begin{cases} i \frac{e}{\sqrt{K^2 + q_z^2}} \sqrt{\frac{\hbar\omega_{\mathbf{K},q_z,\alpha}}{2\epsilon_0 \mathcal{A} L}} \left(\frac{1}{\epsilon_\infty} - \frac{1}{\epsilon_s} \right), & \alpha = \text{LO [7]}, \\ \sqrt{\frac{\hbar\sqrt{K^2 + q_z^2}}{2\rho_m c_{\text{LA}} \mathcal{A} L}} D_{\text{def}}^{c/v}, & \alpha = \text{LA [6]}, \end{cases}\quad (5)$$

i.e. we include one optical mode (LO) in Fröhlich coupling and one acoustic mode (LA) in deformation potential coupling.

Moreover, in Eq. (1), γ_{phon} is the phonon-assisted dephasing, which is obtained by self-consistently solving the following equation:

$$\gamma_{\text{phon}} = \frac{1}{2} \sum_{\mathbf{K}} \Gamma_{\mathbf{K},\mathbf{0}}^{\text{out}} \Big|_{E=\hbar\omega_p, \delta \rightarrow \mathcal{L}\gamma_{\text{phon}}},\quad (6)$$

and $\Gamma_{\mathbf{Q}}^{\text{form}}$ is the phonon-assisted formation rate:

$$\Gamma_{\mathbf{Q}}^{\text{form}} = \Gamma_{\mathbf{0},\mathbf{Q}}^{\text{in}} \Big|_{E=\hbar\omega_p, \delta \rightarrow \mathcal{L}\gamma_{\text{phon}}},\quad (7)$$

where the Dirac delta function in $\Gamma^{\text{out/in}}$ is replaced by a Lorentzian, which takes into account a finite lifetime of higher correlations [8]:

$$\delta(E_{\mathbf{Q}} - \hbar\omega_p \pm \hbar\omega_{\mathbf{Q},q_z,\alpha}) \rightarrow \mathcal{L}_{\gamma_{\text{phon}}}(E_{\mathbf{Q}} - \hbar\omega_p \pm \hbar\omega_{\mathbf{Q},q_z,\alpha}) = \frac{1}{\pi} \frac{\hbar\gamma_{\text{phon}}}{(E_{\mathbf{Q}} - \hbar\omega_p \pm \hbar\omega_{\mathbf{Q},q_z,\alpha})^2 + (\hbar\gamma_{\text{phon}})^2}.\quad (8)$$

Here, ω_p is the center frequency of the optical field.

We note, that we treat the phonons as a bath with a fixed lattice temperature, i.e. the lattice cannot heat up due to optically excited exciton densities within our assumptions. However, we only need an order-of-magnitude estimate of the exciton-phonon scattering times, for which the bath approximation is well sufficient.

We adjust the lattice temperature of the 7.6 nm GaAs quantum well to 6 K, which reflects our measurement conditions.

Via Eq. (6), we calculate a phonon-assisted (half) linewidth of $\hbar\gamma_{\text{phon}} = 8.3 \mu\text{eV}$ at a temperature of $T = 6 \text{ K}$, which is roughly in line with literature values from theory and experiment [9, 10]. Here, we already observe, that exciton-phonon scattering has a negligible impact on the total measured (half) nonradiative linewidth of 0.9 meV.

In Fig. 2(left), we depict the total outscattering rates $\sum_{\mathbf{K}} \Gamma_{\mathbf{K},\mathbf{Q}}^{\text{out}}$ from Eq. (2) from an initial state at \mathbf{Q} integrated over all possible final states \mathbf{K} . In Fig. 2(right), we depict the inverse, the scattering time. We observe, that the scattering time of the interaction of an exciton with LA phonons is of the order of 10-40 ps, much longer than our experimentally observed timeframe of a few ps. The scattering time of the interaction with LO phonons, on the other side, is of the order of 1-2 ps, which, at a first glance, could be relevant within our experimentally observed timeframe.

Figure 2: Calculated outscattering rate (left) and outscattering time (right) of a 7.6 nm GaAs QW at 6 K from Eq. (2).

To examine this issue closer, we depict the exciton dynamics in Fig. 3. Here, the first row displays the dynamics of the coherent ($|P|^2$), incoherent ($N = \frac{1}{A} \sum_{\mathbf{Q}} N_{\mathbf{Q}}$) and total ($|P|^2 + N$) exciton density by taking into account incoherent exciton formation via exciton-phonon scattering "phon", via optical interaction "opt" and via the combined action of exciton-phonon scattering and optical interaction "opt + phon". The second row displays the respective \mathbf{Q} -dependent distributions of the incoherent occupation $N_{\mathbf{Q}}$ at various times. We observe, that only a small fraction of the coherently excited exciton density $|P|^2$ is converted into incoherent exciton densities N via exciton-phonon interaction, cf. "phon" in the first row of Fig. 3, since the radiative and non-radiative dephasing unrelated to phonons is much stronger than the phonon-assisted dephasing. Therefore, a major fraction of the coherently excited exciton density $|P|^2$ is lost before it is converted into incoherent exciton densities N . If we consider the formation via optical interaction, the behavior is very different: Here, a very efficient conversion of coherent to incoherent exciton densities occurs, cf. "opt" in the first row of Fig. 3. If we take into account both mechanisms simultaneously, cf. "opt + phon" in the first row of Fig. 3, nothing changes compared to taking into account optical formation only in "opt". Therefore, phonon-assisted scattering is insignificant within the first few picoseconds. By considering the snapshots in the second row, the reason becomes clear: The optically excited distribution with respect to the center-of-mass momentum of the excitonic occupations $N_{\mathbf{Q}}$ does not extend to the momentum range, where scattering via optical phonons becomes relevant (0.4-0.5 nm⁻¹), cf. Fig. 2, so that scattering with acoustic phonons is the only possibility. This process, as already mentioned, is very slow compared to our ultrafast timeframe of a few picoseconds in our measurements. On a longer timescale of around 120 ps, this process is responsible for the thermalization of the excitonic occupation $N_{\mathbf{Q}}$, where all excess energy due to optical excitation is transferred to the lattice, as depicted by the blue solid lines in "phon" and "opt + phon" in the second row in Fig. 3.

These simulations confirm that under our excitation conditions of ultrafast timescales (few picosec-

Figure 3: First row: Dynamics of the coherent ($|P|^2$), incoherent ($N = \frac{1}{\mathcal{A}} \sum_{\mathbf{Q}} N_{\mathbf{Q}}$) and total exciton density ($|P|^2 + N$) of a single GaAs QW for resonant excitation with a π -pulse of 1345 fs FWHM. Second row: Snapshots of the incoherent exciton occupation $N_{\mathbf{Q}}$ at various times. In both rows, "phon" denotes, that we include only phonon-assisted formation of incoherent occupations, "opt" denotes, that we consider only optical formation of incoherent occupations, and "opt + phon" denotes the inclusion of both mechanisms.

onds) at cryogenic temperatures (6 K), exciton-phonon scattering does not affect the Rabi splitting.

Report of the Second Referee:

This manuscript presents a compelling comparative study of Rabi splitting dynamics in two paradigmatic 2D systems—MoSe2 monolayers and (Ga,In)As quantum wells. The work combines rigorous experiments with a sophisticated microscopic many-body theory, offering novel insights into the interplay between Coulomb interactions and light-matter coupling. The findings are of broad interest to the fields of quantum optics, condensed matter physics, and nanophotonics. This work makes a significant contribution to understanding light-matter interactions in 2D systems. Nonetheless, the following points should be addressed in full:

Our Response:

We are truly grateful for the referee's overall extremely positive statements and appreciation of our work in the thoughtful and encouraging review. We are especially pleased that the referee recognizes the novelty and broader relevance of our comparative study and its contribution to understanding ultrafast light-matter interactions in 2D systems. The clear appreciation of both the experimental rigor and the sophistication of our microscopic theory is deeply motivating. We thank the referee for recognizing the strengths of our work. We have carefully addressed each of their valuable suggestions below in order to improve the manuscript further.

Referee 2:

1. Why the splitting of MoSe2 monolayer is asymmetric? Please provide more detailed discussions.

Our Response:

We thank the referee for addressing this question, which also improved the discussion in the manuscript.

First of all, the overall splitting into a repulsive and an attractive branch reflects the Stark or Rabi splitting induced by the exciting field, i.e., it reflects the light-induced dressed states. To understand the asymmetry of the splitting, we provide three calculations:

(i) In Fig. 4(left), we depict simulations, where we neglected the Coulomb renormalizations and biexcitonic effects, i.e., we take only the first line in Eq. (6) in the SM with optical blocking into account. Here, a fully symmetric Rabi splitting is obtained.

(ii) Now, if we turn on the exciton-density-induced Coulomb renormalizations, cf. the second line in Eq. (6) in the SM, a significant blue shift (repulsive behavior) of the upper branch occurs, as soon as coherent $|P|^2$ or incoherent N exciton densities are present, cf. Fig. 4(middle). The spectral position of the lower branch remains relatively stable, but its strength decreases. We note, that the strong density-dependent blue shift of the upper branch is a common feature of nonlinear absorption measurements in TMDCs [11, 12, 13], while the observation of a lower branch and, hence, a splitting is more challenging due to the necessity of small linewidths.

(iii) If we finally turn on also the biexcitonic effects in the third and fourth line in Eq. (6) in the SM, which result in the formation of the bound biexciton/exciton-biexciton, excitation-induced dephasing and red-shift contributions, cf. Fig. 4(right), the strong density-dependent repulsive behavior of the upper branch is slightly attenuated and accompanied by a broadening due to the biexcitonic continuum, but still exhibits a strong density-dependent overall blue shift.

Therefore, we conclude, that the upper branch is dominated by the strong density-dependent Coulomb interaction, while the lower branch is not. Hence, the latter is a more light-dominated state compared to the former. Thus, in a MoSe₂ monolayer, the initially symmetric light-induced dressing occurring without Coulomb interaction, is significantly altered by the strong Coulomb interaction, which dominates over the optical interaction ($\frac{\hbar\Omega}{E_b} \sim 10^{-3}$), resulting in a strongly asymmetric Rabi splitting both in theory and experiment.

Figure 4: Simulated pump-probe absorption of a h-BN-encapsulated MoSe₂ monolayer at resonant pumping. The grey-shaded area denotes the pump intensity. Left: We take only the first line in Eq. (6) in the SM into account, i.e., we neglect density-dependent Coulomb renormalization and biexcitons/exciton-biexcitons. Middle: We additionally take Coulomb renormalizations in the second line in Eq. (6) in the SM into account. Right: Full simulations, where we additionally take biexcitons and exciton-biexcitons into account as in the main manuscript, cf. lines three and four in Eq. (6) in the SM.

Referee 2:

2. The low-energy branch in the MoSe₂ monolayer simulation is almost missing, which shows a clear deviation with the experimental results.

Our Response:

We thank the referee for rising this issue. However, we would like to emphasize that we found a clear low-energy branch in the simulations, even if it is small. This low-energy branch is almost stationary, in the simulations as well as in the measurements (within uncertainty) and the main excitation-induced shift occurs in the upper-energy branch. The low-energy branch in the simulations, however, disappears more rapidly compared to the experimental results. This is tentatively attributed to the following reason: The MoSe₂ monolayer exhibits some degree of disorder within the illumination area which is not included in the theory. The light-induced state in the experiment might be more susceptible to disorder compared to the Coulomb blue-shifted upper state. Test simulations, where we neglected temporal coherence in the second-order excitonic occupation grating $N^{1,-1}$, cf. Eq. (29) in the supplementary, which causes the overall splitting render a more prominent lower energy peak better suited to the measurements, cf. Fig. 5. At the same time, however, the upper branch deviates more from the experimental observation.

Figure 5: Rabi splitting of a MoSe₂ monolayer. Left: With temporal coherence of the occupation grating as in the main manuscript. Right: Without temporal coherence of the occupation grating.

Since the effect of disorder on and the actual degree of spatial and temporal coherence of the excitonic occupation grating is generally not well understood, differences between the disorder-free simulations and the actual measurements are well within expectation. Yet, in principle, a rigorous treatment of disorder could be implemented within our many-body approach, but is beyond the scope of this manuscript.

Referee 2:

3. What are the changes of MoSe₂ monolayer time evolution of the exciton resonance splitting under different excitation energy densities, and whether this splitting is adjustable? I suggest that the authors should provide more information about this.

Our Response:

We thank the referee for this insightful question, which touches on a central aspect of our study. Indeed, the Rabi splitting in the MoSe₂ monolayer is fluency-dependent and exhibits characteristic nonlinear behavior due to the strong Coulomb interaction in this material system. We now discuss this point in more detail in the revised manuscript, specifically in the context of Fig. 3.

Our experimental data reveal that the energy splitting in the MoSe₂ monolayer increases with excitation fluence; however, this increase is distinctly sublinear unlike the quantum well case. This sublinear behavior stems from strong Coulomb interactions, which dominate over light-matter coupling in this system and significantly alter the nature of the splitting dynamics.

We have incorporated a theoretical analysis based on a Hartree-Fock framework that yields an analytical expression for the Rabi splitting (Eq. (6) in the manuscript), which explicitly captures this

sublinear dependence on the Rabi energy. The model attributes the nonlinearity to the presence of spin-unlike exciton-to-biexciton transitions and the resulting intervalley Coulomb correlations, which dynamically couple the K and K' valleys and lead to an attenuation of the Rabi splitting.

Moreover, the microscopic theory shows that the observed splitting arises not from a polarization grating but from an exciton occupation grating, a process that becomes dominant shortly after the initial excitation. We also observe linewidth narrowing of both split peaks with increasing fluency, further supporting this interpretation. These effects confirm the tunability of the splitting, albeit in a nontrivial, interaction-dominated manner.

To clarify this behavior, we have revised the discussion surrounding Fig. 3 in the manuscript, emphasizing the fluency dependence of the splitting and its many-body origins.

Referee 2:

4. Why do (Ga, In) As quantum wells use 3D false-color representation, which is different from 2D false-color plots of MoSe₂ monolayer? I think it may be more intuitive to use the same representation method for comparison.

Our Response:

We thank the referee for this thoughtful and constructive comment. Our choice of using different visual formats was motivated by the specific features we aimed to emphasize in each material system.

For the (Ga,In)As quantum wells, we opted for a 3D false-color representation to better highlight subtle yet important spectral features—specifically, the Rabi oscillations and transient gain structures. These features manifest as fine modulations in intensity that can be difficult to discern in 2D plots due to limited color resolution. The 3D perspective provides an additional visual dimension, making these modulations more immediately recognizable as variations in amplitude.

In contrast, for the MoSe₂ monolayer, the dynamics are dominated by spectral shifts and splitting without clear Rabi oscillations, and these are well represented in standard 2D false-color maps.

That said, we fully agree with the referee that consistent representations are valuable for direct comparison. To address this, we have added a side-by-side comparison using 2D false-color plots for both materials at two different excitation fluences in Figure 10 of the Supplemental Information.

Referee 2:

5. Can the Rabi oscillations be observed in the transient spectra of MoSe₂ monolayer, like those of (Ga, In) As quantum wells?

Our Response:

We thank the referee for this important question. We carefully analyzed the transient spectra of the MoSe₂ monolayer by integrating the differential transmission over the energy range of the $1s$ exciton resonance (1.6247–1.6513 eV) in Figure 6. While the signal shows a pronounced bleaching at time zero—consistent with phase-space filling—we do not observe any clear signatures of Rabi oscillations in the temporal evolution. Minor variations in signal amplitude at negative delays are attributed to slight changes in probe positioning, resulting in local absorption differences rather than physical dynamics. Beyond the initial pump-induced response, the signal lacks coherent oscillatory behavior, and any residual fluctuations appear random and within the noise level. This absence of Rabi oscillations is consistent with the dominant role of many-body Coulomb interactions and rapid dephasing in the MoSe₂ system, as discussed in the main text.

Figure 6: Time-resolved differential transmission signals of the MoSe₂ monolayer, integrated from 1.6247 eV to 1.6513 eV. The data are shown as a function of delay time around the pump pulse arrival (0 ps). The bleaching at 0 ps indicate the onset of coherent interactions, but no Rabi oscillations are resolved in the transient dynamics. Variations in the pre-pulse signal levels are due to spatial inhomogeneities across different sample positions.

Referee 2:

6. Although the authors have made a detailed theoretical explanation for the Rabi oscillations in the experimental results of (Ga, In) As quantum wells, is there any way to rule out the possibility of phonon oscillations?

Our Response:

We appreciate the reviewer’s thoughtful question. In our case, the possibility of phonon oscillations can be effectively ruled out based on several considerations. The oscillations are directly linked to the resonantly excited 1s exciton transition and exhibit a clear dependence on the excitation field strength, following the expected square-root scaling of the Rabi energy. Here, the oscillation frequency increases with increasing excitation field strength, which is not a property of coherent phonon oscillations. Our microscopic many-body model, which does not include coherent phonon coupling, accurately reproduces the experimental results—indicating that phonons are not necessary to explain the observed behavior. Moreover, the spectra lack phonon sidebands or other features typically associated with coherent lattice vibrations.

To examine coherent phonon oscillations in more detail, we can establish the equations of motion for the coherent phonon amplitude $D_{\mathbf{K},\alpha} = \langle b_{\mathbf{K},\alpha} \rangle + \langle b_{-\mathbf{K},\alpha}^\dagger \rangle$ [14] in the excitonic picture [15], where $b_{\mathbf{K},\alpha}^{(\dagger)}$ are the phonon annihilation (creation) operators at phonon momentum \mathbf{K} and phonon mode α :

$$\partial_t^2 D_{\mathbf{K},\alpha} + \omega_{\mathbf{K},\alpha}^2 D_{\mathbf{K},\alpha} = -\frac{1}{\hbar} \omega_{\mathbf{K},\alpha} \delta_{\mathbf{K},\mathbf{0}} 2\text{Re} \left(g_{-\mathbf{K},\alpha}^c - g_{-\mathbf{K},\alpha}^v \right) |P|^2. \quad (9)$$

Here, $\omega_{\mathbf{K},\alpha}$ is the frequency of the oscillating coherent phonon amplitude, $g_{\mathbf{K},\alpha}^{c/v}$ is the electron-phonon conduction/valence band interaction matrix element, and P is the coherent excitonic transition (we neglect any incoherent occupations), which acts as a source term. Since the optical field strikes the sample perpendicularly, only excitonic transitions at zero center-of-mass momentum $\mathbf{Q} = \mathbf{0}$ are induced. This translates to the fact, that coherent phonons are also only induced at zero momentum $\mathbf{K} = \mathbf{0}$, which is encoded in the Kronecker delta on the right-hand side of Eq. (9). Since it holds: $\omega_{\mathbf{K}=\mathbf{0},\alpha} = 0$, if $\alpha = \text{LA}$, and $\omega_{\mathbf{K}=\mathbf{0},\alpha} = \frac{0.0368}{\hbar}$ eV, if $\alpha = \text{LO}$, only optical phonons oscillate at $\mathbf{K} = \mathbf{0}$. The corresponding period is calculated as: $T = \frac{2\pi}{\omega_{\mathbf{K}=\mathbf{0},\text{LO}}} \approx 112$ fs, with optical phonon energy $\hbar\omega_{\mathbf{K}=\mathbf{0},\text{LO}} = 36.8$ meV [16]. Assuming $P(t) = i\Omega$ (we work in a rotating frame with CW excitation and neglect any coherence decay, Coulomb renormalization and Pauli-blocking), where $\Omega = \frac{d^{cv}E}{\hbar}$ is the Rabi frequency with transition dipole moment d^{cv} and exciting optical field E , we can solve Eq. (9)

as:

$$D_{\mathbf{k},\text{LO}}(t) = -\delta_{\mathbf{k},\mathbf{0}} \frac{2}{\hbar} \text{Re} \left(g_{-\mathbf{k},\text{LO}}^c - g_{-\mathbf{k},\text{LO}}^v \right) \frac{\Omega^2}{\omega_{\mathbf{k},\text{LO}}} \left(1 - \cos(\omega_{\mathbf{k},\text{LO}} t) \right). \quad (10)$$

In comparison, we establish the equations of motion for the coherent exciton density $|P|^2$ in a rotating frame by explicitly taking Pauli-blocking into account but neglecting coherence decay, Coulomb interaction and all contributions $\mathcal{O}(P^4)$ for simplicity:

$$\partial_t^2 |P|^2 + 4\Omega^2 |P|^2 = 2\Omega^2. \quad (11)$$

A solution can be obtained as:

$$|P|^2(t) = \frac{1}{2} \left(1 - \cos(\Omega t) \right), \quad (12)$$

which describes Rabi oscillations [17].

Comparing Eq. (10) and Eq. (12), we observe, that, first, the oscillation frequency $\omega_{\mathbf{k},\text{LO}}$ of the phonon amplitude $D_{\mathbf{k},\text{LO}}$ does not depend on the exciting field strength, while the oscillation frequency Ω of the coherent exciton density $|P|^2$ does, which mirrors the behavior of our measured oscillations. Second, the oscillation period of the phonon amplitude with $T \approx 112$ fs (see above) is much faster than the observed oscillations in our experiment. Therefore, we rule out coherent phonons as a possible source of the measured oscillations.

Referee 2:

7. Why does the authors use such a wide pulse width (1345fs) laser as a pump source for measurement?

Our Response:

The extended pump pulse duration is a direct consequence of our use of spectrally narrow excitation, which was deliberately chosen to achieve precise and resonant excitation of the excitonic transitions. To ensure spectral selectivity, we employed spectral filtering—via a pulse shaper for the (Ga,In)As quantum well system and narrowband spectral filters for the MoSe₂ monolayer. As governed by the time-bandwidth relationship, this spectral narrowing leads to a corresponding temporal broadening of the pulse, resulting in the observed pulse durations. These pulse parameters were carefully optimized for observing clear Rabi splitting in both systems, as discussed in the revised manuscript (cf. Fig. 9 in the SM).

Referee 2:

8. I wonder why the authors use two different lasers for measuring, as it seems inappropriate to compare the results obtained under different experimental conditions.

Our Response:

We appreciate the referee’s concern and welcome the opportunity to clarify our experimental choices. The use of two different laser systems was driven by the specific technical requirements of each sample, particularly in terms of generating stable white-light probes and achieving spectrally narrow excitation at the respective exciton resonances. For the MoSe₂ monolayer, which produces significantly weaker signals and much faster decoherence times, it was essential to operate at a high repetition rate (1030 nm laser system) to ensure sufficient signal-to-noise in the white-light detection. Moreover, a stable white-light continuum required the fundamental wavelength to be sufficiently detuned from the probe spectral window, which further justified the choice of laser. In contrast, the (Ga,In)As quantum well system features narrower spectral resonances. To achieve fine spectral control over the excitation conditions in this system, we used a pulse shaper that allows flexible spectral shaping beyond simple filtering. This level of control required higher pulse energies, which were only available from our 5 kHz amplifier system. Importantly, both laser systems were carefully calibrated to match experimental parameters—such as repetition rate, pulse energy, spectral width, and photon flux—to avoid any artifacts such as sample heating. In fact, we also performed comparative measurements of the (Ga,In)As quantum wells using the 1030 nm system and obtained consistent results, confirming that the observed phenomena are not dependent on the specific laser system but solely on the optical parameters.

All in all, each laser system was chosen to best meet the physical and technical demands of the respective material, and the validity of our comparative analysis is ensured by strict control and cross-validation of all relevant parameters.

References

- [1] Archana Raja et al. “Dielectric disorder in two-dimensional materials”. In: *Nature Nanotechnology* 14.9 (Aug. 2019), pp. 832–837. ISSN: 1748-3395.
- [2] Aidan. P. Rooney et al. “Observing Imperfection in Atomic Interfaces for van der Waals Heterostructures”. In: *Nano Letters* 17.9 (Aug. 2017), pp. 5222–5228. ISSN: 1530-6992.
- [3] Achint Jain et al. “Minimizing residues and strain in 2D materials transferred from PDMS”. In: *Nanotechnology* 29.26 (May 2018), p. 265203. ISSN: 1361-6528.
- [4] Andres Castellanos-Gomez et al. “Deterministic transfer of two-dimensional materials by all-dry viscoelastic stamping”. In: *2D Materials* 1.1 (Apr. 2014), p. 011002. ISSN: 2053-1583.
- [5] Jagdeep Shah, ed. *Hot Carriers in Semiconductor Nanostructures*. Elsevier, 1992. ISBN: 978-0-12-638140-5.
- [6] A Thränhardt et al. “Quantum theory of phonon-assisted exciton formation and luminescence in semiconductor quantum wells”. In: *Physical Review B* 62.4 (2000), p. 2706.
- [7] I-K Oh et al. “Exciton formation assisted by LO phonons in quantum wells”. In: *Physical Review B* 62.3 (2000), p. 2045.
- [8] Frank Lengers, Tilmann Kuhn, and DE Reiter. “Theory of the absorption line shape in monolayers of transition metal dichalcogenides”. In: *Physical Review B* 101.15 (2020), p. 155304.
- [9] S Rudin and TL Reinecke. “Effects of exciton–acoustic-phonon scattering on optical line shapes and exciton dephasing in semiconductors and semiconductor quantum wells”. In: *Physical Review B* 66.8 (2002), p. 085314.
- [10] S Rudin and TL Reinecke. “Exciton-acoustic-phonon linewidths in GaAs bulk and quantum wells”. In: *Physical Review B* 65.12 (2002), p. 121311.
- [11] Chiara Trovatiello et al. “Disentangling many-body effects in the coherent optical response of 2D semiconductors”. In: *Nano Letters* 22.13 (2022), pp. 5322–5329.
- [12] Aleksander Rodek et al. “Local field effects in ultrafast light–matter interaction measured by pump-probe spectroscopy of monolayer MoSe₂”. In: *Nanophotonics* 10.10 (2021), pp. 2717–2728.
- [13] Paul D Cunningham et al. “Resonant optical Stark effect in monolayer WS₂”. In: *Nature communications* 10.1 (2019), p. 5539.
- [14] Alex V Kuznetsov and Christopher J Stanton. “Theory of coherent phonon oscillations in semiconductors”. In: *Physical review letters* 73.24 (1994), p. 3243.
- [15] Florian Katsch et al. “Theory of exciton–exciton interactions in monolayer transition metal dichalcogenides”. In: *physica status solidi (b)* 255.12 (2018), p. 1800185.
- [16] S Rudin and TL Reinecke. “Electron–LO-phonon scattering rates in semiconductor quantum wells”. In: *Physical Review B* 41.11 (1990), p. 7713.
- [17] A Knorr et al. “Asymptotic analytic solution for Rabi oscillations in a system of weakly excited excitons”. In: *Physical Review B* 49.19 (1994), p. 14024.